# Chitosan Nanogel with Mixed Food Plants and Its Relation to Blood Glucose in Type 2 Diabetes: A Systematic and Meta-Analysis Review of Observational Studies

**DOI:** 10.3390/nu14224710

**Published:** 2022-11-08

**Authors:** Morris Aloysius, Kyriacos N. Felekkis, Christos Petrou, Dimitrios Papandreou, Eleni Andreou

**Affiliations:** 1Department of Life and Health Sciences, University of Nicosia, 46 Makedonitissas Ave., CY-2417, P.O. Box 24005, Nicosia CY-1700, Cyprus; 2Department of Health Sciences, College of Natural and Health Sciences, Zayed University, Khalifa B City, Abu Dhabi 144534, United Arab Emirates

**Keywords:** T2D, unripe plantain, bitter yam, okra, chitosan, case-controlled study

## Abstract

This systematic review with metanalysis evaluated and analyzed the beneficial effects of certain plants food in type 2 diabetes (T2D) when consumed alone or in combination with chitosan. The main objective of the paper was to examine the relation of chitosan nanogel and mixed food plant (MFP) to control T2D. The databases included Medline, Scopus, PubMed, as well as Cochrane available between the month of January 1990 to January 2021. The eligibility criteria for selecting studies were case-controlled studies that included unripe plantain, bitter yam, okra, and chitosan either used-alone or in combination with non-specified food plants (NSFP). Two-fold autonomous critics retrieved the information required and evaluated the risk of bias of involved studies. Random-effect meta-analyses on blood glucose controls, were performed. Results of 18 studies included: seven that examined unripe plantains, one bitter yam, two okras, and eight chitosan, found regarding the decrease in blood glucose level. Meta-analysis of the results found a large proportion of *I*^2^ values for all studies (98%), meaning heterogeneity. As a consequence, the combined effect sizes were not useful. Instead, prediction interval (PI) was used (mean difference 4.4 mg/dL, 95% PI −6.65 to 15.50 and mean difference 3.4 mg/dL, 95% PI −23.65 to 30.50) rather than the estimate of its confidence interval (CI). These studies were at 50% high risk of bias and 50% low risk of bias and there was judged to be an unclear risk of bias due to the insufficient information from the included study protocol (moderately low). The intervention lasted between three and 84 days, indicating potency and effectiveness of the intervention at both short and long durations. Due to the moderately low quality of the studies, the findings were cautiously interpreted. In conclusion, the current evidence available from the study does support the relation of chitosan with mixed unripe plantain, bitter yam and okra for the management of T2D. Further high-quality case-controlled animal studies are required to substantiate if indeed chitosan nanogel should be cross-linked with the specified food plant (SFP) for the management T2D.

## 1. Introduction

Hyperglycemia that leads to T2D appears to represent a persistent set of life-sustaining chemical conditions which is described by elevated blood sugar which manifests well as the inadequate and ineffective release of pancreatic hormone and affects a set of life-sustaining chemicals, such as glucose amino acids and fatty acids [1]. Data show that in 2015 more than 400 million adults globally were affected by this condition. These people were predominantly among the economically disadvantaged population, and T2D is projected to be the seventh leading cause of death by 2030 [1]. T2D in contrast to type 1 diabetes (T1D) does not completely rely on insulin. T2D is a metabolic condition largely described for its role in licking the cell membrane, low release of the pancreatic hormone-insulin, as well as the upsurge of blood sugar upon directly ingesting food [1,2].

In third world countries and globally, the predominance of T2D is ascribed to the modifications in lifestyle, including the transition from home prepared food high in phytonutrients (i.e., Polyphenols) to a more Westernized type of food which might be considered less nutritionally sound [3]. Such lifestyle changes have resulted in the increase of the prevalence of prolonged and deteriorating diseases.

The therapeutic benefits of customary foods have assumed significant place lately as a result of benefits connected to their phytonutrients [4,5]. Legumes as well as other food plants possess significant benefits in preventing as well as managing prolonged diseases [4]. Table 1 presents the characteristics of the most frequent drugs used for T2D. 

Synthetic medications, e.g., acarbose, viglibose, and miglitol, are alfa-glucosidase inhibitors while others include biguanides, thiazolidinediones, sulphonylureas, and meglitinides, which display discomforting side effects, such as stomach pain, swelling, the release of ammonia gas from digestion, swelling of the stomach caused by dwelling gas, as well as the loss of fluid via the elementary canal [9]. Some of these discomforting symptoms are perhaps triggered by the production of ethanol as a result of the increased presence of bacteria; an end product from non-catabolized carbohydrates in the gastro intestinal (GI) track. Foods from plant sources generally comprise natural antioxidants such as phenolic compounds that can scavenge for ROS also referred to as free radicals [11,12,13,14,15,16,17]. 

With the use of new oral antidiabetics (OAD), such as gliptines, glucagon-like peptide-1 (GLP-1) analogues, gliflozines were approved for the treatment of T2D, demonstrating improved glycemic control, weight loss, and cardiovascular benefit [7]. The fruit content of plantain (Musa paradisiaca) is considered a major food product in Africa which provides a great source of energy for these people. Plantains are stated as an essential basis for pro-retinoids around continents (Asia, Africa, as well as Latin America). 

*Musa parasidiaca* is rich in fat-soluble retinoids, water soluble B vitamins (thiamin, niacin, riboflavin and pyridoxine), and vitamin C. This food is discovered to be extremely rich in potassium but poor in sodium. *Musa parasidiaca* is an excellent source for vit A compared to other foods. Fat-soluble vit A (carotenoid) is considered to a safeguard against diabetes, heart disease, and cancer. The fat-soluble vitamin is among the best essential groups of phytonutrients which exhibit a vital role in the nutritious value of the plants. Plantain has been used in Nigeria to decrease blood sugar levels after a carbohydrate (CHO) meal. Plantain fits into the family of flowering plants eaten ripe or unripe using diverse methods of processing techniques, being cooked as well as fried. Additionally, products, such as flour and chips, have been made from plantain [10].

Bitter yam (Dioscorea dumetorum) is a food commonly used in tropical countries. This short (small) tuber fits into Dioscorea family. Collectively, it is referred to as bitter yam trifoliate (three–leaved) yam. It is believed to promote glucose balance for the diabetics and to be a therapeutic remedy for different diseases [18,19] Moreover, in the middle-belt region of Nigeria, the Tiv speaking tribe refers to it as ‘Anube’ used for consumption and medicinal purposes to treat illnesses [19]. 

Okra is unique as a flowering plant of the mallow family [13]. The okra fruit is eaten as a common plant in several nations (including Nigeria and Cyprus). It is rich in nutrients. Okra is known for its therapeutic importance, particularly in respect to lowering the blood glucose effect. While okra is usually seen as a plant that is beneficial to diabetic patients, a limited number of technical articles have acknowledged the therapeutic function that okra performs. Earlier research of Sabitha showed that okra oil extracts promoted hypoglycemic and low-fat beneficial effects, as well as improved the weight in Streptozotocin (STZ)-induced diabetic rats. Having a better anti oxidation capability, okra is known to reduce the oxidation of lipids, as well as raise the amount of Superoide Dismutase (SOD), Chloramphenicol acetyltransferase (CAT), as well as Glutathione (GSH). It is worth mentioning that decreased levels of GSH were found in the diabetic rats [9]. 

A useful food can be defined as food which delivers both nutritional and physiological supports to organisms or decreases the possibility of protracted ailments [16]. Okra fruit [17] has previously been shown to establish lower blood sugar as well as lower lipid actions, becoming a versatile and remarkable substitute to control hyperglycemia. 

Nanoparticle production procedures can be easily performed as well as valid amidst wide variable medications. In these systems of delivering medication, polymeric nanoparticles have increased their significance, being eco-friendly, biocompatible, and due to their process of preparation much broadly obtainable. Hence, the variety of uses has been increasing to comprise a multiplicity of chemical medication groups and quantity systems [20,21]. Chitosan-based nanoparticles are mostly suitable and less toxic. Chitosan nanogel has been applied to manage production and the spreading of cell in the body, gastrointestinal tract disorders (GIT) disorder, heart disorder, and channeling medication reaching the central nervous system as well as eye impurities. New investigations in nanogel for oral medication (pills, capsules, syrups) have been centered around premises that increased acceptability of nanogel characteristics as well as techniques involving biochemical changes can be useful in specific nanogel therapeutic production as well as distribution structures. Chitosan is considered as one of the key derived products of chitin, made by eliminating the acetate part from chitin. It is a derivative of crustacean shells, such as those from prawns or crabs, and cell walls of organisms such as fungi. Its occurrence is natural as a polysaccharide. As a cation, extremely basic in nature, Chitin is obtained naturally, linked with peptides as well as elements that require detachment before the preparation of chitosan; hence, the methods of acidifying and alkalizing. After purification, acetyl groups are removed from chitin and substituted with amino group to produce chitosan. Nanogel acts in diffusing the openings of tight junctions of epithelium, enhancing the healing of wounds. Chitosan eases both the transport of therapies between and through cells. Chitosan relates with secretions which are negatively charged to bring about complex hydrophobic (water-hating) interrelationships. The acidic content in the primary amino group of nanogel is 6.5, similar to the amount of acetyl free linked amino group. Likewise, this class of acids aids the dissolving of nanogel hydrogen ion systems as well as the incomplete deactivation of primary amines, which could possibly elucidate the reason why chitosan has been described as concentrating acid from neutral to high hydrogen ion concentration [22]. Therefore, the user of nanoparticles is required to cautiously bring together the preferred chemical and physical characteristics of the chitosan, as well as the expected bio-system, using the chitosan treatment technique.

With this scientific evidence regarding the specified plant food (SFP), unripe plantain, bitter yam, and okra so far, the need to use them for the management of T2D has been limited to combining them alone [3,15,23,24,25] or in combination with NSFP [2,5,26,27,28], and those used separately or in combination are not cross-linked with chitosan [7,10,29]. However, where there is a cross-linkage with chitosan, those linkages are not with the SFP [30,31,32,33] (Table 2).

Chitosan acts by diffusing in the openings of tight junctions of epithelium, thereby enhancing the healing of wounds. The selected food plants (MFP-2 unripe plantain, bitter yam, and okra) acting in conjunction release plant insulin, which ultimately benefits diabetics most especially in complicated states of unhealing wounds on the limbs (the arms and legs). No scientific study so far has identified the combining or mixing effects of SFP with chitosan for the management of T2D. Hence, the present systematic and meta-analysis review aims to examine the relation of chitosan nanogel and MFP in T2D. MFP could improve the blood glucose level in association with antidiabetic drugs [10]. 

## 2. Methods

The systematic review was conducted in agreement with the 2009 PRISMA statement [43]. The review procedure was recorded with PROSPERO in March 2019 (CRD 42019129124).

### 2.1. Search Strategy

We searched for journal papers indexed in PubMed, Medline, Scopus, as well as Cochrane that were available from January 1990 to January 2021. The search was limited to T2D alone, generally available in the English language using the search terms: “unripe plantain, bitter yam, okra, chitosan with T2D.’’

### 2.2. Study Selection, Inclusion as Well as Exclusion Criteria

The review included only studies that examined unripe plantain, bitter yam, okra, and chitosan and their relation to type 2DM. Additionally, no human studies were included. Furthermore, we set a 28-year exploration boundary since therapeutic designs of over 30 years-old may change significantly from natural plant therapy to nutraceutical patterns [15]. Articles were searched and adjudicated. All searched articles were screened by titles and abstracts so as to retrieve useful journals deemed to be eligible. We revised journal paper completely according to standards set. Forms of differences were resolved by consensus-oriented discussion; where there was no resolution, a third party was consulted. 

### 2.3. Information Extraction

Information on features involved in the revisions was taken out individually with the help of two critics. The first critic extracted textual data and the other graphical data (Figures). Authors were contacted where necessary through their e-mails or phone numbers published in their articles for additional or missing data. The data extracted were related to the number of experimental groups in the study design (i.e., the group which was induced and the one receiving intervention). Both species and sex of animal related to the characteristics of the animal model were extracted from the data. The intervention of interest, dosage, timing of dosage, and effectiveness of dosage on the animal’s data was extracted [44]. The primary outcome data support a reduction in hyperglycemia and the unit of measurement was milligrams per deciliter (mg/dL) or millimole per litre (mmol/L). All extracted data were in a continuous data pattern.

### 2.4. Assessing the Risk Bias

The method of assessing bias risk or assessing quality involved individually assessing risk associated with bias as well as studies included, which were evaluated by two external reviewers, one on text data and the other on graphical data (Figures). Disagreement was resolved by consensus-oriented discussions and where resolution was not reached, a third party was consulted. Studies were evaluated with the Systematic Review Centre for Laboratory Animal Experimentation (SYRCLE) tool for assessing bias risk in animal studies [37]. We regarded individual areas to be at ‘low risk’, ‘unclear risk’ or ‘high risk’ of bias. We generally categorized the bias risk to be ‘low’ when the answer to the signaling questions for that domain was “Yes”, ‘high’ when the answer to the signaling questions was “No”, or as ‘unclear’ when there was insufficient information about that domain (Table 3).

### 2.5. Strategy for Information Investigation

Effects regarding outcome remained stated with statistical difference at 95%CI, considered from both end standards or modifying the starting point. Through studies, effects on blood glucose level remained constantly obtainable as milligram per deciliter or millimole per liter (mg/dL or mmol/L). 

We synthesized approximations in statistical terms, employing prototypical unsystematic effects meta-analysis, founded on the postulation that comparable as well as procedural heterogeneity remained probable to occur as well as have consequence on the outcomes [45,46]. We employed regression on the moderator effect to evaluate differences among studies, as well as considered 95% CI employing moderator analysis (technique of moment estimation) (Figure 1).

Arithmetical irregularity remained calculated employing *I*^2^ value (Table 4 and Table 5). 

We generated funnel plots (Figure 2) to examine the minor effect-change from studies (affinity about mediation properties projected in lower investigations varies since the ones projected in higher investigations may affect publication preferences, protocol, and comparable differences, among influences). Evaluations were directed employing meta-essentials software intended for meta-analyzing investigations and the result was a change in the statistics among free sets [45].

Hooijmans [37] reported that calculating the instant mark among separate investigations was not a good practice using the SYRCLE’s tool because an instant mark will involve allocating “loads” particular to known areas to tool, which will be problematic when justifying loads allocated. Moreover, loads could vary for every result and for every analysis. 

### 2.6. Patients as Well as General Participation

While the investigation limited the enrollment of human participants, the evaluation was entirely focused on animals only. However, due to the nature of the research question, its result was extrapolated to patients and the public.

## 3. Outcomes (Results)

### 3.1. Explored Outcomes

The exploration of four automated records (Medline, Scopus, PubMed, and Cochrane) identified 405,919 collections through 454 journal papers left over once excluding replicas. Out of this number, 417 journal papers were disqualified as the titles and abstract were lacking our criteria, and those journal investigation reports were not of the standards set. Out of 37 selected journal papers, 19 investigational journal report papers were dropped due to not been a precise case investigation, in vivo as well as in vitro animal study, not SFP cross-linked with chitosan, not having modulator blockers or enzyme inhibitors, and not insulin stimulated or secreted on account of SFP as outcome. Thus, 18 studies were identified as eligible for inclusion in the review (Figure 3) [43]. 

### 3.2. Features of Involved Investigation as Well as Valuation of Intervention: T2D

The features of the involved investigations are presented in Table 6.

The invesstigations consisted of seven involving unripe plantain [2,5,11,26,35,36,47], one involving bitter yam [25], two involving okra [15,24], and eight involving chitosan [7,10,29,30,31,32,33,48] respectively. Almost all the included studies were in vivo case-controlled studies which were induced with diabetes using streptozotocin (STZ) and four others with alloxan monohydrate [2,5,30,35]. The major intervention from the studies was Type 2 diabetes with few other interventions in some studies, such as enzyme inhibition, weight change, lipid decrease/insulin resistance, sustained release time, and neuropathy. In all these interventions, SFP were used either in combination with other (NSFP) or used alone without cross-linking them with chitosan. As such, chitosan was used alone to encapsulate either insulin or other active compounds to aid their release over a sustained time. The main outcome measure in all the included studies was a decrease of blood glucose level in the animals.

### 3.3. The Bias Risk across Investigations

Complete information of the risk bias investigation of animal studies is presented in Table 7.

Amongst 18 case precise studies, bias risk was shared among the included studies between high risk of bias and low risk of bias for items present for the SYRCLE tool [44]. This high or great bias risk was due to the allocation concealment, unsystematic housing, blinding, and incomplete outcome data. This is a common practice in animal studies as the current design of protocols and reporting of animal studies are very poor [37].

### 3.4. Decrease in Blood Glucose Level

Hypoglycemic measures were examined in six studies [3,5,15,36,40,42]. Each of the studies were further examined by separating them into experimental groups: negative diabetic control at baseline and diabetic intervention group (the former was given no intervention-SFP or NSFP); subgroups were involved in meta-analysis as well as practical statistically significant positive effect on the studies (Figure 4). Therefore, the combined effect sizes of the studies were (mean difference 4.0 mg/dL, 95% CI −0.33–8.40 as well as the difference in mean of 4.4 mg/dL, 95% CI −0.82–9.66) as shown in the Forest plot (Figure 4 and Figure 5 respectively.

Due to the large proportion of *I*^2^ value (98%) in the studies (Table 4), we explored both the subgroup (Figure 4) and moderator (Figure 5) analyses, which proved that the investigations for meta-analysis came from a heterogeneous population. Hence, the extent of heterogeneity was examined. The random effect model was used to assume that there was heterogeneity in the subgroups. Consequently, the combined effect sizes in the subgroups (AA and BB) were not used. Instead, the prediction interval was used (mean difference 4.4 mg/dL, 95% prediction interval (PI) −6.65 to 15.50 and mean difference 3.4 mg/dL, 95% prediction interval −23.65 to 30.50) [45,49].

Due to the large proportion of *I*^2^ value (98%) in the studies (Table 4), we explored both the subgroup (Figure 4) and moderator (Figure 5) analyses, which proved that the investigations for meta-analysis came from a heterogeneous population. Hence, the extent of heterogeneity was examined [45,49]. The random effect model was used to assume that there was heterogeneity in the subgroups. Consequently, the combined effect sizes in the subgroups (AA and BB) were not used. Instead, the prediction interval (PI) was used (mean difference 4.4 mg/dL, 95% (PI) −6.65 to 15.50 and mean difference 3.4 mg/dL, 95% PI −23.65 to 30.50) rather than the estimate of its confidence interval (best vital result of ‘random outcomes’ representative; once the situation needs to be presumed ‘true’, outcome dimensions differ)

Regarding regression on moderator effect size (Figure 4), there was an observable strong correlation among moderator as well as detected influence dimensions. These were long-established with significant outcomes in the importance test of regression load *p* < 0.05 (Table 8).

This proved the value effect of the study sizes on the intervention. Furthermore, Figure 5 presents the funnel plot for the random effect meta-analysis on the mean difference in the decrease of blood glucose (mg/dL) based on the intervention or negative diabetic control, indicating asymmetry in the distribution of the effect sizes. Therefore, studies of this effect (X→Y) have been conducted in the following populations. The following weight change and enzyme inhibition population have not been studied. Observed effects range from −0.33 to 8.40. Effects in subgroup A “Diabetic intervention” range from −0.82 to 9.66. (Table 8) [45].

## 4. Discussion

This systematic review of case-controlled animal studies examining the decrease of blood glucose level in diabetic animals found evidence to support the notion that chitosan nanogel can relate to mixed unripe plantain, bitter yam, and okra in controlling T2D. The results remained similar when subgroup analysis was performed. The review questions the single use of the SFP, the use in combination with NSFP, and without cross-linking them with chitosan controlling T2D.

### 4.1. Principal Findings

Meta-analysis of the diabetic intervention subgroup demonstrated decreased hyperglycemia against blood sugar negative regulator [11,15,36,40,42,47]. However, studies from subgroup AA (diabetic intervention) influenced much of the bases for the overall decrease in blood glucose level as these studies were studies which had as primary outcome measure the decrease of blood glucose and secondary outcome measures including increase of body weight, inhibition ofα-amylase and α-glucosidase, as well as neuropathy [11,36,47,49].

Studies of the subgroup included in the meta-analysis gave results that agreed with the different analysis proving heterogeneity of the six studies formed the *I*^2^ value (Forest plot, subgroup analysis, moderator analysis and publication bias analysis [45]). As a consequence of this development, the overall combined effect sizes from the meta-analysis (Forest plot) were not useful due to the non-homogeneous studies included (not from a single population of population samples) [45,49].

### 4.2. Quality of Evidence

We reflected on the value of evidence which was reasonably low due to the following explanations: involved investigation remained at 50% high risk of bias and 50% at low risk of bias; with the high-risk bias coming from the designed protocol [44]. We also saw a high level of heterogeneity among the involved investigations (diabetic intervention). This heterogeneity could reflect the different populations being examined. For instance, the populations of the diabetic interventions that also had weight change, enzyme inhibition, and neuropathy were different from the population of diabetic intervention by a sustained release-time [4,31,41,47].

Ten out of the eighteen studies included in the systematic review were studies based on the SFP and NSFP (coco- yam, soya bean cake, cassava fibre and rice bran) [2,5,47,50], while eight studies used only chitosan to encapsulate active bio-compounds [16,30,33,39,40,41,42].

Furthermore, investigation involved in the review lasted between three and 84 days (three days [10,29]; five days [30]; eight days [30]; 10 days [42]; 14 days [3,32,42]; 21 days [36]; 28 days [3,4,5,24,33,35,51]; 31 days [50], and 84 days [15]). This proved the potency and efficacy of the intervention over both short and long durations. This meant that there was strong effect in the intervention given to the animals.

### 4.3. Limitation

This review had limitations. Our search strategy could have omitted abstracts and full text articles that were published in other languages besides English. This omission could have affected the number of studies included in the meta-analysis and the nature of the result with respect to a more homogeneous population [14,17].

## 5. Conclusions and Future Implication

As the quality of the included studies was moderately low in percentage (50%), which means that the overall risk of bias was unclear risk (50% low risk of bias and 50% high risk of bias), there was insufficient information provided by the study authors in their protocol, and final outcomes ought to be inferred cautiously. However, current evidence available does support the relation of chitosan to mixed unripe plantain, bitter yam, and okra for the management of T2D. We recommend that high quality case-controlled animal studies are carried out to substantiate whether chitosan nanogel should indeed be cross-linked with the SFP for the management of T2D.

Furthermore, research efforts could be geared towards extracting the phytonutrients contained in these food plants, concentrate and fractionate (partitioned) alike, so that extract fractions obtained can be used to test the efficacy of these food plants either through in vitro or in vivo experimentations.

## Figures and Tables

**Figure 1 nutrients-14-04710-f001:**
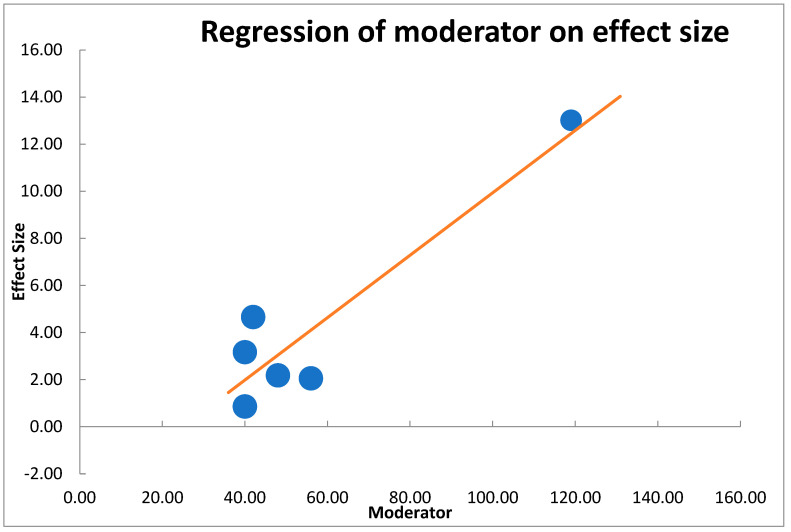
Moderator analysis showing strong correlation between moderator and effect size for the included studies in meta-analysis.

**Figure 2 nutrients-14-04710-f002:**
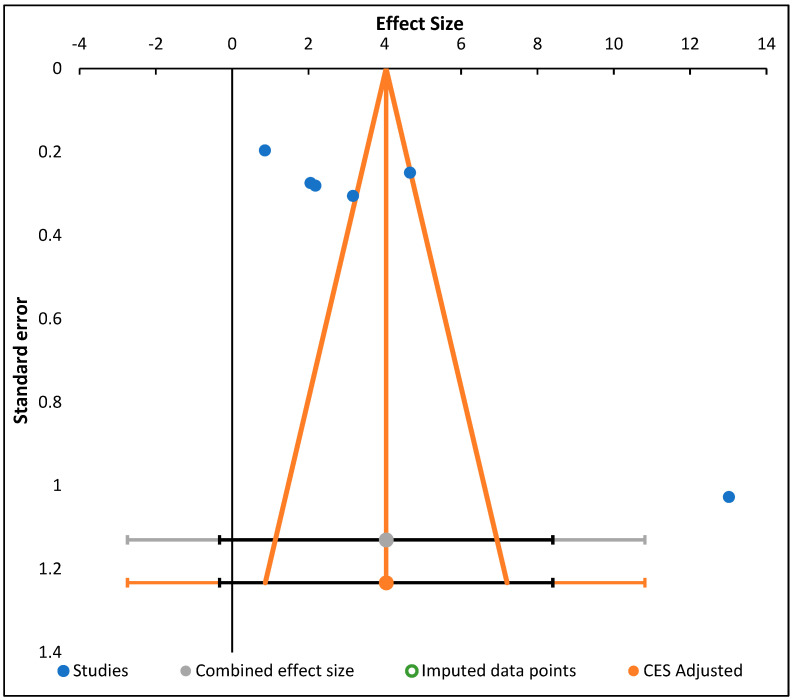
Funnel plot showing random effect meta-analysis of mean difference in decrease of blood glucose level (mg/dL) based on diabetic intervention or negative diabetic control.

**Figure 3 nutrients-14-04710-f003:**
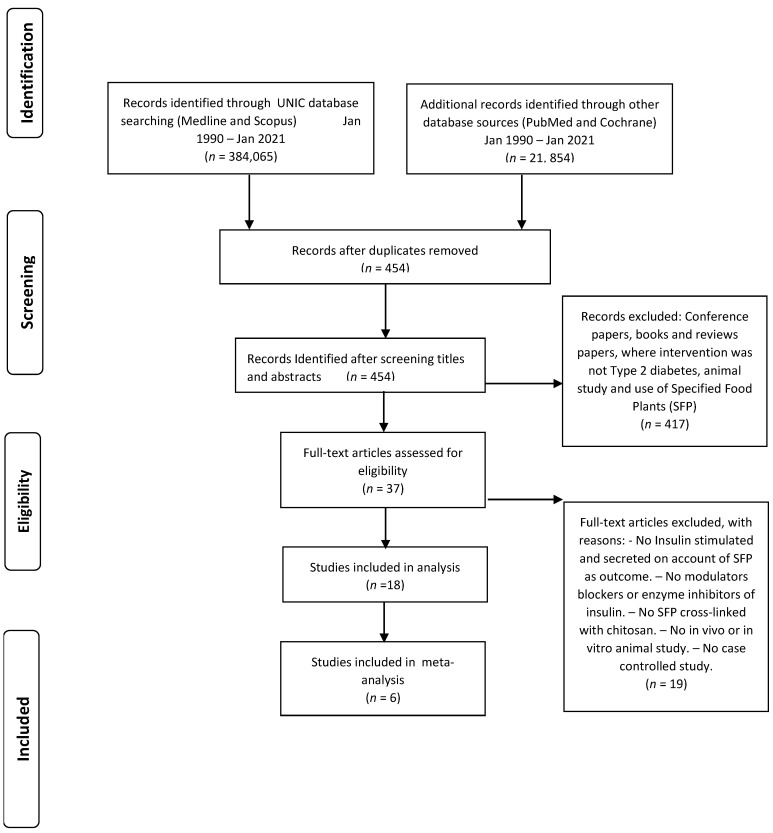
PRISMA flow diagram of included articles.

**Figure 4 nutrients-14-04710-f004:**
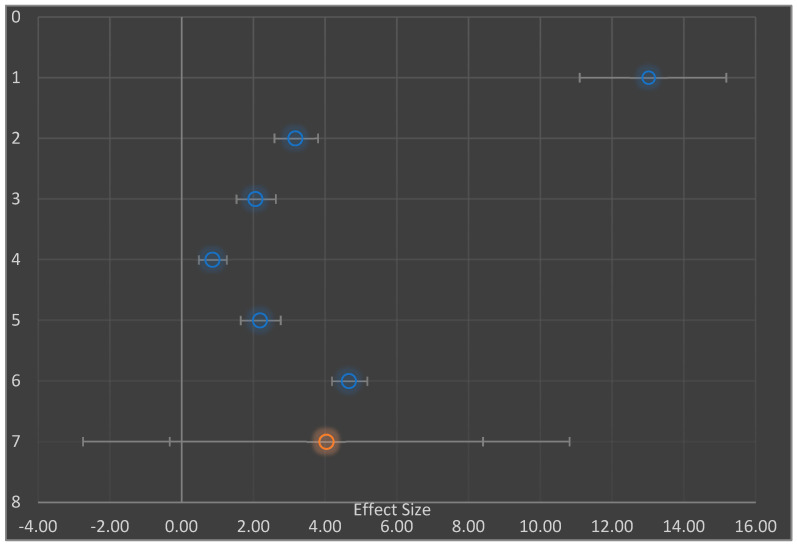
Forest Plot showing random effect meta-analysis of the mean difference in control of blood sugar level (mg/dL), based on diabetic intervention and negative diabetic control. Data for studies 1, 2, 3, and 4 are based primarily on control of blood sugar level whereas data for studies 5 and 6 are based on control of blood sugar level in sustained release-time.

**Figure 5 nutrients-14-04710-f005:**
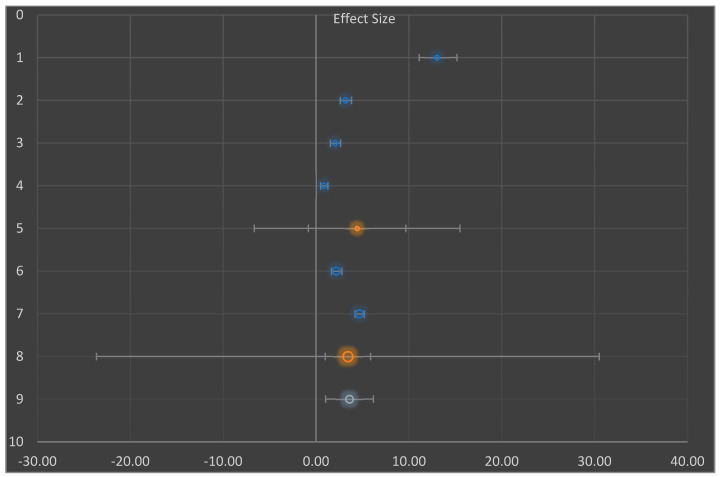
Showing random effect meta-analysis of the mean different in total subgroup analysis for the decrease in blood sugar level (mg/dL).

**Table 1 nutrients-14-04710-t001:** Showing characteristics of common antidiabetic drugs used for T2D treatment.

Class of Antidiabetic Drug	Specific Mechanism of Action	Adverse Effect	Reference
Acabose—Used for treating T2D	Acarbose works by slowing down the action of certain chemicals that break down food to release glucose into the blood; slow food digestion helps to keep blood glucose from rising high after any meal.	Hyperglycemia, Shaking, Dizziness, Sweating, Irritability, Mood change, Headache, Numbness, Weakness, Pale skin, Hunger, Clumsy, Confusion, Seizures, Loss of consciousness, Extreme thirst, Frequent urination, Blurred vision, Dry mouth, Stomach upset, Vomiting, Shortness of breath, Breath that smells fruity and decreased consciousness.	[6]
Voglibose—Used as an α-glucosidase inhibitor that manages postprandial blood sugar in T2D.	α-glucosidase inhibitor; the saccharides, acting as competitive inhibitor of enzymes needed to digest carbohydrate specifically the α–glucosidases enzymes present in brush border of small intestine.	Seen in about 25% of users. Adverse effects include: Soft stool, Diarrhoea, Flatulence, Bloating, Abdominal pain or fullness and nausea.	[7]
Glyset (Miglitol)—A drug employed to treat symptoms of T2D. It can be used alone or in combination with this class of drugs. It belongs to a class of drugs referred to as Antidiabetics, α-glucosidase inhibitors.	Unlike Sulonylureas, Glyset (Miglitol) does not enhance insulin secretion. Antihyperglycemia action of Miglitol results from a reversible inhibition of membrane-bound intestinal α-glucosidase hydrolase enzymes.	Hives, Difficulty breathing, swelling of the face, Lips, Tongue, or throat, Severe diarrhea, Constipation, Bloody or tarry stools, rectal bleeding and diarrhea that contains blood or mucus.	[8]
Biguanides—This refers to a group of oral diabetes drugs that work by preventing the production of sugar in the liver, improving the body sensitivity towards insulin and reducing the amount of sugar absorbed by the intestines.	It works by preventing the liver from converting fat and amino acids into sugar. They also activate an enzyme which helps cells respond more effectively to insulin and takes in sugar from the blood. It is used by obese people as it promotes weight loss.	Hypoglycemia results very rarely, weight gain and digestive adverse reactions.	[9]
Thiazolidinedione (TZDs)—These are insulin sensitizers that act on intercellular metabolic pathways to enhance insulin action and increase sensitivity in critical tissues.	TZDs act by activating peroxisome proliferator-activated receptors. They are also agonist. The endogenous ligands for those receptors which are fatty acids and eicosanoids. This binds DNA when receptors activated.	Increase hepatitis and possible liver failure, Edema, Heart failure, Coronary Heart Disease (CHD), Plaque progression, Myocardia	[9]
Sulfonylureas—Stimulates the release of insulin from pancreatic Beta-cells and have a number of extra-pancreatic effects	Induces sugar independent-insulin release from Beta-cells by inhibiting potassium flux through Adenosine Triphosphate (ATP) dependent potassium channels.	Hypoglycemia, Induces hyponatremia, Edema, Induces alcohol flushing.	[10]
Meglitinides—these are oral drugs used for T2D. They work by triggering production of insulin.	Meglitinide (Rpaglinide)—This is an insulin secretagogue meaning that it binds to receptors on pancreatic beta-cells and stimulates insulin release. Repaglinide binds to an ATP-dependent potassium channel on beta-cells.	Hypoglycemia (low blood sugar) is associated with increased mortality and weight gain	[9]

**Table 2 nutrients-14-04710-t002:** Food plants and chitosan used for their therapeutic and benefit in T2D.

Food Plant and Chitosan	Therapeutic Use	Benefits/Nutritional Value	Reference
Unripe plantain (Musa paradisiaca)	Food plant was used alone. Exert anti-hyperglycemic effects due to the inhibition of α-amylase and α-glucosidase activities by their essential phytochemicals as well as their amylose/amylopectin contents in diabetic rats.	Source of nutrients vital to human well-being i.e., Moisture, Crude protein, Crude lipid, Ash, Crude fiber, Carbohydrate, and gross energy 148.6 Kcal/100 g. Mineral content of sodium, Potassium, Calcium, Magnesium, Iron, Phosphorus, Zinc, Manganese, and Copper	[3,34]
Unripe plantain (Musa paradisiaca); soya bean cake and cassava fibre.	Used in combination.	Moisture, Crude protein, Crude lipid, Ash, Crude fiber, Carbohydrate, and gross energy 148.6 Kcal/100 g. Mineral content of sodium, Potassium, Calcium, Magnesium, Iron, Phosphorus, Zinc, Manganese, and Copper	[23,34]
Unripe plantain (Musa paradisiaca) and (Dioscorea rotundata)	Used in combination. Anti-hyperglycemic and and anti-hyperlipemic effect. The *M. paradisiaca-based* diet significantly (*p* < 0.05) overturned the levels of fasting blood glucose, with significant (*p* < 0.05) increase in insulin and glycogen concentrations. The diet also increased the activity of hexokinase with significant reduction (*p* < 0.05) in glucose-6-phosphatase and fructose-1-6-diphosphatase activities.	Moisture, Crude protein, Crude lipid, Ash, Crude fiber, Carbohydrate, and gross energy 148.6 Kcal/100 g. Mineral content of Sodium, Potassium, Calcium, Magnesium, Iron, Phosphorus, Zinc, Manganese, and Copper. Diocorea rotundata; a family of the Dioscorea contains similar nutritional value such as Flavonoids, Alkaloids, Saponins. Cardiac glycosides are present in small quantities. Tannins and anthraquinones are absent.	[22,34,35]
Unripe plantain (Musa paradisiaca)	Used alone. The hyperglycemic mice needed vital modification (*p* < 0.05) with hyperglycemia. Unripe plantain could play a key role in diabetes management.	Moisture, Crude protein, Crude lipid, Ash, Crude fiber, Carbohydrate, and gross energy 148.6 Kcal/100 g.	[5,34]
Unripe plantain (Musa paradisiaca L.); cocoyam (Colocasia esculenta L.).	Used in combination. Feeding of rats with test feeds for 21 days to the diabetic rats of groups (Grps) 4 and 5, brought about 58.75% and 38.13% reductions in hyperglycemia. Cocoyam and *Musa parasidiaca* can management of hyperglycemic kidney disease.	Moisture, Crude protein, Crude lipid, Ash, Crude fiber, Carbohydrate, and gross energy 148.6 Kcal/100 g.	[34,36]
Unripe plantain (Musa paradisiaca L.); soya bean cake and rice bran.	Used in combination. The dough meals decreased blood glucose by approximately 76% in diabetic rats stabilizing at the same time the lipid profiles to normal physiological range. The combination of plantain, soybean cake and rice bran was able to yield functional dough with good nutritional composition, pasting properties, and antidiabetic potential; decreasing serum glucose of diabetic rats by more than 70%. The formulated functional dough considered promising diet that can be recommended for patients with low economic status suffering from diabetes, especially in Sub-Saharan Africa.	Moisture, Crude protein, Crude lipid, Ash, Crude fiber, Carbohydrate, and gross energy 148.6 Kcal/100 g.	[27,34]
Unripe plantain (Musa paradisiaca L.), Ginger (Zingber *officinale*).	Used in combination with Group iii as well as vi mice needed 159.52 percent as well as 71.83 percent reductions in blood glucose; however, 24.91 percent as well as 35.32 percent reductions regards heaviness related to group I as well as ii mice which needed 2.09 percent as well as 22.94 percent upsurge in blood glucose by 13.42 percent rise with 45.36 percent reductions via weightiness separately. Combination of *Musa parasidiaca* as well as *Zingiber officiale* whose dosage needed for controlling hyperglycemia remained imprecise related with *Musa parasidiaca* unaided.	Moisture, Crude protein, Crude lipid, Ash, Crude fiber, Carbohydrate, and gross energy 148.6 Kcal/100 g	[4,34]
Bitter yam (Dioscorea batatas)	Used alone. The results show that yam and allantoin (another component inside bitter yam) have antidiabetic effects by controlling antioxidant activities, lipid profiles and by promoting the release of GLP-1, thereby improving the function of β-cells maintaining normal insulin and glucose levels.	Diocorea batatas; a family of the Dioscorea contains similar benefit nutritionally such as Flavonoids, Alkaloids, Saponins. Cardiac glycosides are present in small quantities. Tannins and anthraquinones are absent along rich mineral elements.	[22,37]
Okra (Abelmoschus esculentus)	Used alone. Fruits and seeds, but significantly (*p* < 0.05) reduced the total phenolic content as well as their free radical scavenging capacity. Regular administration of processed and untreated fruit (UTF) and seed suspensions significantly reduced (*p* < 0.05) the blood glucose level of rats. Boiling and roasting do not significantly affect the antidiabetic potential of *A. esculentus* fruits and seeds.	Source of nutrients that are vital to human well-being, e.g., vitamins, potassium, calcium, carbohydrates, dietary fiber, and unsaturated fatty acids like linolenic and oleic acids, and similarly of bioactive chemicals.	[24,38]
Okra (Abelmoschus esculentus)	Used alone. Serum glucose sharply increased and insulin resistance was noted in the diabetic rats (glucose: 360–500 mg/dl, Homeostasis model of assessment for insulin resistance (HOMA-IR) 9.8–13.8). F2 preparation, rich in polysaccharides and carbohydrates, was most effective in reducing hyperglycemia and insulin resistance (glucose: 200 mg/dl; HOMA-IR: 5.3) and especially HbA1C (from 8.0% to 6.5%). All of the Okra subfractions reduced the level of triglycerides and free fatty acid, but not the level of total cholesterol. Whether it is consumed as a vegetable or as a nutraceutical, Okra has the potential to be an alternative therapy for diabetes.	Source of nutrients that are vital to human well-being, e.g., vitamins, potassium, calcium, carbohydrates, dietary fiber, and unsaturated fatty acids like linolenic and oleic acids, and similarly of bioactive chemicals.	[15,38]
Chitosan	Used alone without cross-linking the SFP. The palmityl acylated exendin-4 (Ex4-C16) release from deoxycholic acid-modified glycol chitosan (DOCA-GC) nanogels was seriously delayed against Ex4. DOCA-GC nanogels were deposited quickly after pulmonary administration and persisted in the lungs for approximately 72 h. The hypoglycemic period of inhaled Ex4-C16 nanogels was much better than that of Ex4 nanogels in *db*/*db* mice.	Chitosan enhances the healing of wounds. Chitosan also eases both transport of medications between and through cells.	[39]
Chitosan	Used with non-SFP and was cross-linked with extract of the non-SFP. The highest dose levels of *Punica granatum* (Pg) extract, rind extract and its spray dried biopolymeric dispersions with casein (F1) or chitosan (F2) showed notable hypoglycemic activity with 48, 52, and 40% reduction in the mice glucose levels after 6 h, respectively.	Chitosan enhances the healing of wounds. Chitosan also eases both transport of medications between and through cells.	[40]
Chitosan	Used alone without linkage with the SFP. The low molecular weight chitosan (LMWC) LMWC-exendin-4 conjugate formed a nanoparticle structure with a mean particle size of 101 ± 41 nm through complexation between the positively charged LMWC backbone and the negatively charged exendin-4 of individual conjugate molecules. The biological activity of the LMWC-exendin-4 conjugate was evaluated in an INS-1cell line. The absorbed exendin-4 demonstrated a significantly enhanced hypoglycemic effect. The LMWC-exendin-4 conjugate could be used as a potential oral anti-diabetic agent for the treatment of type 2 diabetes.	Chitosan enhances the healing of wounds. Chitosan also eases both transport of medications between and through cells.	[41]
Chitosan	Used alone without cross-linkage with SFP. The oral administration of 50 IU/kg insulin-loaded nanoparticles to type 1 diabetic rats resulted in longer period of anti-hyperglycemic effects up to 12 h and relative pharmacological availability of 5.04% comparing to the subcutaneous administration. The oral anti-hyperglycemic effect was further compared between type 1 and type 2 diabetic models by the intraperitoneal glucose tolerance test, showing that the effect lasted longer in the type 1 diabetic model. The biopolymeric-based delivery. Nano particulate system is a promising tool for the therapy of Type 1 and Type 2 diabetic patients and prevention of Type 1 diabetes.	Chitosan enhances the healing of wounds. Chitosan also eases both transport of medications between and through cells.	[42]
Chitosan	Used without cross-linkage with SFP. The in vitro and in vivo release of Ex4-C16 from Decanoic acid-modified glycol chitosan (DA-GC) hydrogels was intensely delayed compared to native Ex4 possibly due to strong hydrophobic interactions. Ex4-C16 in DA-GC hydrogels was found to be present around the injection site up to 10 days after subcutaneous administration, whereas Ex4 in DA-GC hydrogels was not seen in the injection sites in approximately 2 days in ICR mice. The hypoglycemia induced by Ex4-C16 DA-GC hydrogels was maintained for >7 days. Ex4-C16 DA-GC hydrogels offer a potential delivery system for the long-term effect treatment of Type 2 diabetes.	Chitosan enhances the healing of wounds. Chitosan also eases both transport of medications between and through cells.	[30]
Chitosan	Used without cross-linkage with SFP. Nanolayer cross-linkage and the development of complexes permitted a superior loading capacity of insulin (0.90%), as well as improved stability and 74% reduced solubility at acid pH in vitro, compared to non-cross-linked insulin. The cross-linked insulin administered by oral gavage reduced fasting blood glucose levels by up to 50% in a sustained and dose-dependent manner and reduced postprandial glycemia in streptozotocin-induced diabetic mice without causing hypoglycemia. Nanolayer cross-linkage decreased the likelihood of rapid and erratic falls of blood glucose levels in animals. A capable method to motivate the intestinal absorption efficiency and release performance of the hormone, possibly permitting an efficient and safe route for oral insulin delivery of insulin in diabetes management.	Chitosan enhances the healing of wounds. Chitosan also eases both transport of medications between and through cells.	[31]
Chitosan	Used without cross-linkage with SFP. The in vitro results indicated that all tested samples had comparable rat α-glucosidase inhibitory and porcine α-amylase inhibitory activity. Based on these extrapolations, it was decided to further examine the effect of all three samples at a dose of 0.1 g/kg, on decreasing postprandial blood glucose levels in Sprague-Dawley (SD) rat model after sucrose loading test. In the animal trial, all tested samples had postprandial blood glucose decrease effect, when compared to control.	Chitosan enhances the healing of wounds. Chitosan also eases both transport of medications between and through cells.	[32]
Chitosan	Used without cross-linkage with SFP. Development of microparticles prepared by spray drying of 2% alginate solution encapsulated by 0.1% chitosan was characterized by good mucoadhesive properties, high drug loading and longer period of metformin hydrochloride release. Designed microparticles decreased rat glucose blood level, delayed absorption of metformin hydrochloride and provided stable plasma drug concentration.	Chitosan enhances the healing of wounds. Chitosan also eases both transport of medications between and through cells.	[33]

**Table 3 nutrients-14-04710-t003:** **Use of** SYRCLE’s tool for assessing risk of bias.

Item	Type of Bias	Domain	Description of Domain	Review Author’s Judgement
1	Selection bias	Sequence generation	There is direct evidence that cases and controls were similar, recruited within the same time frame, and controls are described as having no history of the outcome.	Yes *
2	Selection bias	Baseline characteristics	There is direct evidence that appropriate adjustments were made for covariates and confounders in the final analyses through the use of statistical models to reduce research-specific bias including standardization, matching of cases and controls, adjustment in multivariate model, stratification, propensity scoring, or other methods were appropriately justified.	Yes
3	Selection bias	Allocation concealment	There is insufficient information on concealment in the allocation of the animals into groups and subgroups.	No *
4	Performance bias	Random housing	There is direct evidence that the housing of animals was not random as they were kept in cages in the animal house.	No
5	Performance bias	Blinding	There is direct evidence that caregivers and researchers were not blinded or information was not provided.	No
6	Detection bias	Random outcome assessment	There is indirect evidence that it was possible for outcome assessors to infer the exposure level prior to reporting outcomes.	No
7	Detection bias	Blinding	Investigators also served as outcome assessors. There is direct evidence that exposure was consistently assessed using well-established methods that directly measure exposure like the blood glucose levels.	Yes
8	Attrition bias	Incomplete outcome data	There is no information provided on subject removal or exclusion from the study.	No *
9	Reporting bias	Selective outcome reporting	There is direct evidence that all of the study’s criteria were measured in the protocol, such as methods, abstract and introduction have been reported.	Yes *
10	Others	Other sources of bias	There is direct evidence that the other bias like “Units” was reported. Appropriate units such as mg/dL and mmol/L were assigned.	Yes *

* Items in agreement with the items in the Cochrane Risk of Bias tool, Yes = Low Risk of Bias, No = High Risk of Bias.

**Table 4 nutrients-14-04710-t004:** Subgroup meta-analysis showing effect size, *I*^2^ and PI values for AA and BB.

#	Study Name/Subgroup Name	Effect Size	CI Lower Limit	CI Upper Limit	Weight	Q	P_Q_	*I* ^2^	T^2^	T	PI Lower Limit	PI Upper Limit
1	Shodehinde et al., 2015 [3]	13.02	10.98	15.06	21.82%							
2	Eleazu and Okafor; 2015 [5]	3.17	2.56	3.78	25.94%							
3	Eleazu et al., 2013 [36]	2.06	1.51	2.60	26.03%							
4	Huang et al., 2017 [15]	0.86	0.47	1.25	26.22%							
5	AA	4.42	−0.82	9.66	17.78%	162.77	0.000	98.16%	5.01	2.24	−6.65	15.50
6	Lopes et al., 2017 [42]	2.18	1.62	2.74	49.87%							
7	Raafat and Samy, 2014 [40]	4.66	4.17	5.15	50.13%							
8	BB	3.43	0.99	5.86	82.22%	43.54	0.000	97.70%	3.00	1.73	−23.65	30.50
9	Combined Effect Size	3.60	1.03	6.17		258.29	0.000	98.06%	4.07	2.02	1.03	6.17

N.B, AA = Diabetic intervention group, BB = Negative diabetic control group.

**Table 5 nutrients-14-04710-t005:** Publication bias analysis confirming effect sizes and *I*^2^ values.

Combined Effect Size	Observed	Heterogeneity	
Effect Size	4.03	Q	258.29
Standard error	1.70	P_Q_	0.000
CI Lower limit	−0.33	*I* ^2^	98.06%
CI Upper limit	8.40	T^2^	4.07
PI Lower limit	−2.75	T	2.02
PI Upper limit	10.82		
**Combined effect size**	**Adjusted**	**Trim and Fill**	On
Effect Size	4.03	Estimator for missing studies	Leftmost Run/Rightmost run
Standard error	1.70	Search from mean	Left
CI Lower limit	−0.33	Number of missing studies	0
CI Upper limit	8.40		
PI Lower limit	−2.75		
PI Upper limit	10.82		

**Table 6 nutrients-14-04710-t006:** General characteristics of included studies Baseline characteristics of rats or cell lines.

Stydy/Country	N, Characteristics [No. of Cells-In Vitro or No of Rats-In Vivo, Weight (Wt.) of Rats (Rts) No of Group (Grp)]	Study Duration	FP or Material and Family/BM/T	Reagent for Induction of Diabetes/SL(Average) Post Induction	Description of the Study	Outcome Measures with the Use of FP
Shodehinde, S.A. et al. Life Sci. 2015 [3]/ Nigeria	42 male rats (in vivo), Wt = 200 g, Grp-7	14 days	Musa parasidiaca (Musaaceae)/Carotenoids, Polyphenols, Potassium and Vitamin C/NT	Streptozotocin/≥250 mg/dL	The effect of the diets on the blood glucose level, pancreatic α-amylase, intestinal and α-glucosidase content of the unripe plantain products was determined.	Blood glucose level from 350 to 200 mM/L (*p* < 0.05), α-amylose and α-glucosidase (*p* < 0.05).
Famakin, O. et al. J. Food Sci. Technol. 2016 [23]/Nigeria	60 Wistar albino, Wt = 150 g Grp-6	28 days	Musa parasidiaca (Musaaceae)/ Carotenoids, Polyphenols, potassium and Vitamin C/NT	Alloxan monohydrate/≥250 mg/dL	Four of the Grps were fed with prepared food samples (PSC-1, PSC-2, PSC-3 and 100 % plantain). The remaining Grps were treated with Cerolina (a control sample) and metformin hydrochloride (an antidiabetic drug) and commercial animal feed.	Blood glucose change from 355 ± 43 to 103 ± 14 mg/dL (*p* < 0.05). Weight change noted
Ajiboye, B.O. et al., Food Sci Nutr. 2018 [35] Nigeria	48 Albino rat, Wt = 150 ± 20 gGrp- 4	28 days	Musa parasidiaca (Musaaceae)/Carotenoids, Polyphenols, potassium and Vitamin C/NT	Alloxan monohydrate/≥250 mg/dL	Grp 3 diabetic rats fed with D. rotundata-based diet and administered metformin orally (14. 2 mg/kg) and Grp 4 diabetic rats fed with unripe M. paradisiaca-based diet. Grp 1 non-diabetic rats fed with D. rotundata flour-based diet. Grp 2 diabetic control rats fed with D. rotundata-based diet.	Blood glucose change from 350 to 100 mg/dL (*p* < 0.05). Physique heaviness as well as hyperglycemia was measured
Eleazu, C.O.; Okafor, P. Interv. Med. Appl. Sci. 2015 [5] Nigeria	48 male albino, Wt = 243 g, Grp-3	28 days	Musa parasidiaca (Musaaceae)/Carotenoids, Polyphenols, potassium and Vitamin C/NT	Streptozotocin/≥200 mg/dL	Grp 3 diabetic rats fed with unripe plantain pellets (81%) Grp 1 normal rats fed with standard rat pellets (non-diabetic control) Grp 2 diabetic control rats fed with standard rat pellets Blood test was monitored for Rts nourished with *Musa parasidiaca* related by non-diabetic rats.	Hyperglycemia as well as stopping blood test for diagnosing or monitoring hyperglycemic problem. Ratio of % hyperglycemia 2.05 to −159.52% (*p* < 0.05).
Eleazu, C.O.; et al. J. Diabetes Res. 2013 [36] Nigeria	40 male albino, Wt= 288.74 g	21 days	Colocasia esculenta and Musa parasidiaca (Musaaceae)/Carotenoids, Polyphenols, potassium and Vitamin C/NT	Streptozotocin/≥200 mg/dL	Grp 4 diabetic rats administered cocoyam combined feed; Grp 5 diabetic rats administered unripe plantain combined feed. Grp1 normal rats administered standard rat pellets (non- diabetic control); Grp 2 diabetic control rats administered 55 mg/kg body weight STZ; Grp 3 diabetic control rats administered 70 mg/kg body weight STZ	Hyperglycemia as well as heaviness change Decreased of SL by 38.13% (*p* < 0.05) *Musa parasidiaca* decreased Wt and growth by 29.52%
Sukanya, C. et al. Glyset (Miglitol). Bull Environ Contam Toxicol. 2016 [27]. Nigeria	35 Wistar albino, Wt = 150 g, Grp-5.	28 days	Musa parasidiaca (Musaaceae); Glycine max (Fabaceae/Carotenoids, Polyphenols, potassium and Vitamin C-/NT-	Alloxan monohydrate, ≥250 mg/dL,	Three groups were fed with the experimental diets (PSR1, PSR2, and PSR3), Another group was treated with metformin hydrochloride (antidiabetic drug) and saline solution and fed with commercially available animal feeds, and the last group was fed with Cerolina (a control flour sample commonly recommended for diabetic patient)	Lowered blood glucosefrom 129.92 ± 52.80 to 55.00 ± 12.25 mg/dL (*p* < 0.05) Stabilizing the lipid profiles
Iroaganachi, M.; et al., Biochem. J. 2015 [4]. Nigeria	30 male albino, Wt = 232.91 g, Grp-4	28 days	Musa parasidiaca (Musaaceae); *Zingber officinale* (Zingiberaceae)/Carotenoids, Polyphenols, potassium and Vitamin C/NT	Streptozotocin, ≥200 mg/dL	Grp 3. Diabetic rats fed unripe plantain combined feeds (810 g/kg) Grp 4 diabetic rats fed unripe plantain and ginger combined feeds (710: 100 g/kg) Grp 1 normal rats fed with standard rat pellets (non-diabetic control). Grp 2 diabetic control rats which also received standard pellets I	Blood glucose change from 252.25 ± 40.32 to 97.20 ± 11.10 mg/dL (*p* < 0.05). Weigh change noted. 45.36 % reduction of heaviness
Hooijmans, C.R.;et al., ILAR J. (2014), [37]. Nimenibo-Uadia, R. et al., Pa. J. Appl. SCI. Environ Manag. 2017 [22]. Republic of Korea	50 male Sprague-Dawley (SD), Wt = 250 g, Grp-5	31 days	Bitter yam (Dioscorea batatas)/ Polyphenols and Alkaloids, Tannins and anthraquinones are absent/None T.	Streptozotocin, 350 mg/dL	Suspended crude yam powder-treated diabetic; water extract of yam-treated diabetic and allantoin-treated diabetic group normal control, STZ-induced diabetic control	Blood glucose change from 494 ± 81 to 128 ± 20 mg/dL (*p* < 0.05). Blood lipid change noted.
Nguekouo, P.T.;et al., J. Food Biochem. 2018 [24]. Thule, Umpierrez. Curr Diab Rep, 2014, 14 [28]. Cameroon	60 wistar albino, Wt = 200 g, Grp-6	28 days	Okra-Abelmoschus esculentus (Mallow) Malvaceae/ Polysaccharides and major flavonoids, isoquercitrin and quercetin 3-O-gentiobioside, Different levels of Asenic is reported to absorbed into the plant/None T.	Streptozotocin, ≥200 mg/dL	Grps 1 and 2 served as untreated fruits and boiled fruits (200 mg/kg)-treated diabetic rats, respectively. Grps 3 and 4 served as untreated seeds and roasted seeds (200 mg/kg)-treated diabetic rats, respectively. Grp 6 served as diabetic control (positive) received distilled water (4 ml/kg). The animals fed with normal diet were used as normal control (negative) Grp and received distilled water (4 mL/kg) and Grp 5 served as standard drug, metformin (300 mg/kg)-treated diabetic rats	Blood glucose change from 333 ± 20 to 119 ± 18 mg/dl (*p* < 0.05)
Huang, C.; et al., PLoS One. 2017 [15]. South Africa	Different levels of Asenic is reported to absorbed into the plant 32 male Sprague-Dawley, Wt = 250 ± 20 g Grp10.	84 days	Okra (*Abelmoschus esculentus*)/Polysaccharides and major flavonoids, isoquercitrin and quercetin 3-O-gentiobioside/NT.	Streptozotocin, ≥250 mg/dL	Diabetes with 0.23 mg/kg F1, diabetes with 0.45 mg/kg F1, diabetes with 0.23 mg/kg F2, diabetes with 0.45 mg/kg F2, diabetes with 0.23 mg/kg FR, diabetes with 0.45 mg/kg F3. Control (normal diet), C1-C3 (normal diet with 0.45 mg/kg F1, F2, or FR added), HFD with STZ injection; diabetes model Serum glucose quickly increased and insulin resistance was noted in the diabetic rats (glucose: 360–500 mg/dl, HOMA-IR 9.8–13.8). F2, rich in polysaccharides and carbohydrates, was most effective in reducing hyperglycemia and insulin resistance (glucose: 200 mg/dl; HOMA-IR: 5.3)	Blood Glucose change from 500 to 120 mg/dL Insulin resistance noted.
Lee, J.; et al., J. Control Release. 2012 [39]. Republic of Korea	48 male *db*/*db* mice, Wt unspecified, Grp-8,	3 days	Chitosan (Poliglusan, Deacetylchitin; Poly-D)glycosamines), Chitopharm/ Natural biopolymer; obtained via deacetylation process of chitin/NT	Insufflation Administration, Blood glucose confirmed by glucometer at specified time-range	Pulmonary hypoglycemic efficacies, 50 μL of Ex4 or Ex4-C16 DOCA-GC nanogels (320 μg/mL as Ex4 or Ex4-C16; 100 nmol/kg mouse, and 20 mg/mL as DOCA-GC) were administered by insufflation to male *db*/*db* mice A portion (50 μL) of Ex4 or Ex4-C16 solution (100 nmol/kg) was administered as positive controls, and Ex4 non-loaded DOCA-GC nanogels were also administered as a negative control. The lung deposition of DOCA-GC nanogels was monitored using an infrared imaging system, and the hypoglycemia caused by Ex4-C16-loaded DOCA-GC nanogels was evaluated after pulmonary administration in T2D *db*/*db* mice. the Ex4-C16 release from DOCA-GC nanogels was seriously delayed vs. Ex4. DOCA-GC nanogels were deposited quickly after pulmonary administration and stayed in the lungs for approximately72 h. Also, the hypoglycemic period of inhaled Ex4-C16 nanogels was much better than that of Ex4 nanogels in *db*/*db* mice.	Blood Glucose change from 150 to 57 mg/dL (*p* < 0.05) Hypoglycemia and drug sustained release time.
Raafat, K.; et al., Evid. Based Compl. Alt. 2014 [40]. Lebanon	112 Swiss webster mice, Wt = 280 gGrp-16.	8 days	Pomegranate (*Punica granatum* L.) Lythraceae (Punicaceae), Natural biopolymer; obtained via deacetylation process of chitin/NT	Alloxan monohydrate, 200 mg/dL	Grps 3, 4, and 5 was given the Pg extract dissolved in vehicle at doses of 25, 50, and 100 mg/kg, respectively. (iv) Grps 6, 7, and 8 was given F1, suspended in vehicle at doses equal to 25, 50, and 100 mg Pg extract/kg, respectively. (v) Grp 8 was given placebo F1 suspended in vehicle (200 mg/kg). (vi) Grps 9, 10, and 12 was given F2, suspended in vehicle at doses equal to 25, 50, and 100 mg Pg extract/kg, respectively. (vii) Grp 13 was given placebo F2 suspended in vehicle (200 mg/kg). (viii) Grps 15, 16, and 17 was given gallic acid (GA) dissolved in DMSO, at doses of 3,6, and 12 mg/kg to the animals, respectively. Grp 1 was given only vehicle (0.9 % sterile saline) and served as control. (ii) Grp 2 was given glibeclimide dissolved in DMSO as reference drug (5 mg/kg).	Blood glucose change from 210.15 ± 7.30 to 138.89 ± 1.45 mg/dL (*p* < 0.05) and DNnoted.
	112 Swiss, webster mice, Wt= 280 g, Grp-16	8 days	Pomegranate (*Punica granatum* L.) Lythraceae (Punicaceae)/Natural biopolymer; obtained via deacetylation process of chitin/NT	Alloxan monohydrate, 200 mg/dL	(iii) Grps 4, 5, and 6 was given Pg extract dissolved in vehicle at doses of 25, 50, and 100 mg/kg, respectively. (iv) Grps 7, 8, and 9 was given F1 suspended in vehicle at doses equal to 25, 50, and 100 mg Pg extract/kg, respectively. (v) Grp 10 was give placebo F1 suspended in vehicle 200 mg/kg. (vi) Grps 11, 12, and 13 was given Pg F2 suspended in vehicle at doses equalt to 25, 50, and 100 mg Pg extract/kg, respectively. (vii) Grp 14 was given placebo F2 suspended in vehicle 200 mg/kg. (viii) Grps 15, 16, and 17 was given GA, dissolved in DMSO, at doses of 3,6, and 12 mg/kg, respectively. Grp 2 was given only vehicle (0.9% sterile saline) and served as diabetic control. (ii) Grp 3 was given glibeclimide dissolved in DMSO as reference drug (5 mg/kg, IP). Subacute (8 days) effect of various doses of Pg, F1, and F2 and the active compounds.	Blood glucose change from 214.21 ± 9.70 to 181.56 ± 2.02 mg/dL (*p* < 0.05) and DN noted.
Ahn, S.; et al., J. Control Release, 2013 [41]. Republic of Korea	Mouse, Pancreatic Cell lines (INS-1), 18 male *db*/*db* mice, Wt unspecified Grp-3)	3 days	Chitosan (poliglusan, Deacetylchitin; Poly-(D)glycosamines), Chitopharm/ Natural biopolymer; obtained via deacetylation process of chitin/NT	Formulation and glucose administered, intraperitoneal glucose tolerance test (IPGTT	Exendin-4-cys, and LMWC-exendin-4 conjugate.PBS control (PBS-EDTA buffer pH = 7.4). The LMWC-exendin-4 conjugate formed a nanoparticle structure with a mean particle size of 101 ± 41 nm through complexation between the positively charged LMWC backbone and the negatively charged exendin-4 of individual conjugate molecules. The biological activity of the LMWC-exendin-4 conjugate was estimated in an INS-1 cell line. The LMWC-exendin-4 conjugate stimulated insulin secretion in a dose dependent manner as similar as that of natural exendin-4.	Insulin concentration and dose release time. Blood glucose change from 211.93 ± 4.50 to 142.44 ± 2.80 mg/dL (*p* < 0.05)
Lopes, M.;et al.; Eur. J. Pharm. Biopharma. 2017 [42]. Portugal	36 male wistar, Wt= 350 g, Grp-3	10 days	Chitosan can also be referred at as poliglusan, Deacetylchitin; Poly-(D)glycosamines, Chitopharm, Natural biopolymer; obtained via deacetylation process of chitin/Non-toxic,	Streptozotocin, ≥14 mM Subcutaneous (S.C.) Administration	50 IU/kg insulin-loaded NP, (ii) empty NP. (iii) without administration, compared with control Wistar rats (*n* = 10). The oral administration of 50 IU/kg insulin-loaded nanoparticles to Type 1 diabetic rats resulted in extended period of antihyperglycemic effects up to 12 h and relative pharmacological availability of 5.04% comparing to the subcutaneous administration.	Sustained drug-release time. Blood Glucose change from 234.21 to 164.56 mg/dL (*p* < 0.05)
	40 male wistar, Wt = 350 gGrp-5	10 days	Chitosan can also be referred at as poliglusan, Deacetylchitin; Poly-(D)glycosamines, Chitopharm, Natural biopolymer; obtained via deacetylation process of chitin.	Streptozotocin, ≥14 mM, intraperitoneal glucose tolerance test (IPGTT).	50 IU/kg insulin-loaded NP, (ii) 100 IU/kg insulin-loaded NP or (iii) non-encapsulated insulin, all orally delivered by gavage. (iv) subcutaneous (s.c.) non-encapsulated insulin. The oral antihyperglycemic effect was further compared between T1D T2D by the intraperitoneal glucose tolerance test, revealing that the effect lasted longer in the T1D.	Sustained drug-release time 10 days
Lee, C; et al., Acta Biomater. 2014 [30]. Republic of Korea	Murine, melanoma B16F10 cell lines (20,000 cells seeded), 18 male *db*/*db* mice, Grp-3 Wt not specified	5 days	Chitosan (poliglusan, Deacetylchitin; Poly-(D)glycosamines), Chitopharm/Natural biopolymer; obtained via deacetylation process of chitin/NT	Subcutaneous (S.C.) Administration.	Hypoglycemic efficacies, Ex4 or Ex4-C16 DA-GC hydrogels (100 u.g as Ex4) were administered by single s.c. injection using a 21-gauge needle into male *db*/*db* mice Blank DA-GC hydrogels were also administered as controls. The in vitro and in vivo release of Ex4-C16 from DA-GC hydrogels was effectively delayed compared with natural Ex4 probably due to strong hydrophobic interactions. Specifically, Ex4-C16 in DA-GC hydrogels was found to be present around the injection site up to 10 days after subcutaneous administration, whereas Ex4 in DA-GC hydrogels was cleared from injection sites in approximately2 days in ICR mice.	Sustained drug release-time; hypoglycemia effect. Blood Glucose change from 291.83 to 144.44 mg/dL (*p* < 0.05)
Song, L.; et al., Int. J. Nanomedicine. 2014 [31]. United Kindom	30 male mice, Wt= 30 g Grp-5	14 days	Chitosan can also be referred at as poliglusan, Deacetylchitin; Poly-(D)glycosamines, Chitopharm, Natural biopolymer; obtained via deacetylation process of chitin/NT	Streptozotocin, ≥≥20 mmol/L	The resulting preparations were administered separately to the diabetic mice with doses of 30 and 60 units/kg (fasting) and 60 and 120 units/kg (fed): multilayer-coated capsule containing insulin microparticles, multilayer-coated capsule containing insulin-chitosan particles (20 IU/kg). The following preparations were administered separately to the diabetic mice with doses of 30 and 60 units/kg (fasting) and 60 and 120 units/kg (fed): insulin-alone microparticles, insulin-chitosan microparticles, and ip injection of the free-form insulin solution (20 IU/kg).	Blood glucose change from 60 to 20 mMolL (*p* < 0.05) and sustained drug release dosage.
Jo, S.; et al., Int. J. Mol. Sci. 2013 [32]. Republic of Korea	30 male Sprague Dawley (SD), Wt= 200 g, Grp-6	14 days	Chitosan (poliglusan, Deacetylchitin; Poly-(D)glycosamines), Chitopharm/ Natural biopolymer; obtained via deacetylation process of chitin/NT,	Sucrose and inhibitors administered, Diabetes confirmed by glucose oxidase method	Inhibitors (GO2KA1, GO2KA2, GO2KA3) administered. No inhibitors and Acarbose administered. Entirely verified models needed up surges in hyperglycemia decreasing outcome, once related toward regulating, conversely the supplementation needed highest outcome.	Blood Glucose change From193 to 152 mg/dL (*p* < 0.5), α-amylase and α-glucosidase inhibition at sustained time.
Szekalska, M.; et al., 2017 [33]. Poland	30 male Sprague Dawley (SD), Wt = 250 ± 20 g Grp-5.	28 days	Chitosan can also be referred at as poliglusan, Deacetylchitin; Poly-(D)glycosamines, Chitopharm, Natural biopolymer; obtained via deacetylation process of chitin/NT	Streptozotocin, ≥250 mg/dL	Non cross-linked microparticles preparation C, cross-linked microparticles preparation CH1M and commercially available tablet with metformin hydrochloride (MF) Rats treated with carboxymethyl cellulose sodium salt (Control), microparticles placebo (ALG).	Blood glucose change from 305 to 262 mg/dL (*p* < 0.05) and drug release time.

Grp—Group; Wt—Weight; Rts—Rats; FP—Foof Plant; BM—Bioactive Molecules; T—Toxicity; STZ—Streptozotocin; SL—Sugar level; NT—Non-Toxic; DN—Diabetic Neuropathy.

**Table 7 nutrients-14-04710-t007:** Risk of bias assessment in case-controlled animal studies.

Selection Bias	Selection Bias	Selection Bias	Performance Bias	Performance Bias	Detection Bias	Detection Bias	Attrition Bias	Reporting Bias	Others	Overall Review Author’s Judgement	Reference
Low risk	Low risk	High risk	High risk	High risk	High risk	Low risk	High risk	Low risk	Low risk	Unclear *	[3]
Low risk	Low risk	High risk	High risk	High risk	High risk	Low risk	High risk	Low risk	Low risk	Unclear *	[23]
Low risk	Low risk	High risk	High risk	High risk	High risk	Low risk	High risk	Low risk	Low risk	Unclear *	[35]
Low risk	Low risk	High risk	High risk	High risk	High risk	Low risk	High risk	Low risk	Low risk	Unclear *	[5]
Low risk	Low risk	High risk	High risk	High risk	High risk	Low risk	High risk	Low risk	Low risk	Unclear *	[36]
Low risk	Low risk	High risk	High risk	High risk	High risk	Low risk	High risk	Low risk	Low risk	Unclear *	[2]
Low risk	Low risk	High risk	High risk	High risk	High risk	Low risk	High risk	Low risk	Low risk	Unclear *	[4]
Low risk	Low risk	High risk	High risk	High risk	High risk	Low risk	High risk	Low risk	Low risk	Unclear *	[37]
Low risk	Low risk	High risk	High risk	High risk	High risk	Low risk	High risk	Low risk	Low risk	Unclear *	[24]
Low risk	Low risk	High risk	High risk	High risk	High risk	Low risk	High risk	Low risk	Low risk	Unclear *	[15]
Low risk	Low risk	High risk	High risk	High risk	High risk	Low risk	High risk	Low risk	Low risk	Unclear *	[39]
Low risk	Low risk	High risk	High risk	High risk	High risk	Low risk	High risk	Low risk	Low risk	Unclear *	[40]
Low risk	Low risk	High risk	High risk	High risk	High risk	Low risk	High risk	Low risk	Low risk	Unclear *	[41]
Low risk	Low risk	High risk	High risk	High risk	High risk	Low risk	High risk	Low risk	Low risk	Unclear *	[30,42]
Low risk	Low risk	High risk	High risk	High risk	High risk	Low risk	High risk	Low risk	Low risk	Unclear *	
Low risk	Low risk	High risk	High risk	High risk	High risk	Low risk	High risk	Low risk	Low risk	Unclear *	[31]
Low risk	Low risk	High risk	High risk	High risk	High risk	Low risk	High risk	Low risk	Low risk	Unclear *	[32]
Low risk	Low risk	High risk	High risk	High risk	High risk	Low risk	High risk	Low risk	Low risk	Unclear *	[33]

Unclear * = Insufficient information; NB: Refer to Table 3, for better understanding of Table 4.

**Table 8 nutrients-14-04710-t008:** Regression of moderator on effect size showing a model y= −3.31791 + 0.13252x and *p* < 0.05.

	B	SE	CI LL	CI UL	β	Z-Value	*p*-Value
Intercept	−3.31791	1.81	−7.98	1.34		−1.83	0.067
Moderator	0.13252	0.03	0.06	0.21	0.91	4.45	0.000
**Analysis of variance**	**Sum of squares (Q*)**	**df**	** *p* **		**Mean square**	**F-Value**	***p*-value**
Model	19.79	1	0.000		19.79	19.75	0.011
Residual	4.01	4	0.405		1.00		
Total	23.80	5	0.000				
Combined effect size	3.98						
T^2^ (method of moments estimation)	3.34						
R^2^	83.16%

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
