# Peer review of "Chitosan Nanogel with Mixed Food Plants and Its Relation to Blood Glucose in Type 2 Diabetes: A Systematic and Meta-Analysis Review of Observational Studies"

_nutrients, 2022, doi:10.3390/nu14224710_

Round 1

Reviewer 1 Report

Each abbreviation should write the full name in the first time.

Line158-159: Web of Science?

Table 1 and Table 2: please redesign, and further refinement.

Line 419: Author Contributions?

As a review, Future perspectives should be provide in detail. In this study, it was too simple.

Author Response

Hello,  We took your comments into great consideration. The changes are included in the modified article

Regards 

EA

Reviewer 2 Report

Dear authors,

The article submitted by E. Andreou, and their colleagues analyzed the benefits of some plants on diabetes mellitus pathology.

Some issues must be clarified before consideration for a possible publication. Please find my suggestions and comments below:

1.     The title should be more concise

2.     Why do the authors choose chitosan and the mixed food plants (MFP-2 unripe plantain, bitter yam and okra)? This issue should be clearly stated.

3.     Lines 49-52 and 55-59: authors should rephrase these sentences. The actual meaning could be ambiguous.

4.     Actual data regarding the diabetes prevalence should be presented.

5.     Lines 63-64: “As a result, there is increase in the prevalence for protracted as well as deteriorating diseases” – should be reworded

6.     Lines 68-70: Authors should present the classes of antidiabetic drugs and their specific mechanism of action. Also, their adverse effects

7.     Authors must abbreviate the words after their first appearance in the text. Please, check the use of abbreviations in detail

a.      T1D, T2D

b.     Line 108: STZ

c.      Line 122: GIT

d.     Line 145: SFP

e.      Line 153: MFP, etc.

8.     Line 153: MFP might lower the blood sugar, but in my opinion the claim that MFP controls T2D is too much. Please revise this aspect in the entire manuscript.

9.     Authors should conform their manuscript to MDPI recommendations

10.  Table 2: authors must correct “Streptozotocn”

11.  Author Contributions should be presented

12.  More current references should be used

Author Response

Hello

We took into great consideration your comments 

Please, review in the modified article

BR

EA

Round 2

Reviewer 1 Report

ok

Author Response

Thank you for your input. It was taken under consideration and appropriate  amendments were made accordingly 

Reviewer 2 Report

Dear authors,

Please find below my sugestions:

1. Table 1 should be updated:

- acarbose, miglitol, and voglibose are alfa-glucosidase inhibitors.

- you should use only INN not brand names

- biguanides are used in obese patients because mainly they reduce the body weight. Also, hypoglycemia is very rarely.  Digestive adverse reactions are their main risk.

- you should add the newest OAD classes: gliptines, GLP-1 analogues, gliflozines.

2. MFP could improve the glycemia level. They could control the DM in association with antidiabetic drugs. I suggest to reformulate all manuscript. You can use other formulations such as  reducing the blood glucose level, reduce hyperglycemia, etc.

Author Response

Thank you for your comments. Amendments were made accordingly.